# DESIGN FOR TRUSTWORTHY AI SOLUTIONS

## ABSTRACT

Transparency of an AI solution is the need of the hour. With growing adoption, AI is increasingly making business critical decisions in organizations and propagating it, not only limited to organizations but also to society. This has resulted in growing legislative asks from organizations such as "US algorithmic accountability act -2019", "EU right to Explainabilty", "EU AI act 2021" and so on. We here by propose a framework for addressing transparency in AI solution and bringing about trustworthiness, reliability and un-biasness of AI solution to various stakeholders, which may include but not limited to, AI solution engineers, chief legal counsel, decision reviewers etc. Our solution addresses the problem via providing transparency in terms of **Data Interpretation**, where we use AI to spot historical bias, mitigate them and perform risk assessment of the data; **Conformity Assessment**, where we test trustworthiness, robustness and explainability of AI algorithm; **Decision Intelligence**, where we provide insights on financial impact, potential risks and scope for human intervention based on business and regulatory requirements.

## 1 INTRODUCTION

Unconscious and conscious bias has always been rampant in human decisions. Past human decisions are baked into historical data and when used for building AI system results in biased outcomes which could have adverse effects (Hellstrom¨ et al., 2020). In the recent past, many of the AI applications that perform decision making on behalf of humans have come under scrutiny because of their direct impact on society and for not being fair, explainable and transparent (Weber et al., 2020; Zorio, 2021; MEHRABI et al., 2022; Leavy, 2018; Wachter et al., 2020). With growing AI adoption, AI is increasingly making business critical decisions in organizations and so the challenges are growing exponentially. In order to make sure of the impact of high risk AI applications, various legal regulations are coming into picture, which includes but not limited to "US Algorithmic Accountability Act -2019" (116th Congress , 2019), "EU right to Explainabilty" (EUROPEAN COMMISSION, 2021) etc., that mandate AI systems to ensure discrimination is not propagated to further hurt protected demography.

Several approaches have been proposed in recent times to deal with the problem of biasness from data perspective in detecting and mitigating biasness and evaluation of treatment on protected attributes (Dwork & Ilvento, 2018; Kearns et al., 2017; Pastor et al., 2010; Wiśniewski & Biecek, 2022; Jiang & Nachum, 2022). Also from the algorithmic stand point, many approaches have been proposed to deal with algorithmic biasness (Mehrabi et al., 2022; Agarwal et al., 2018; Roselli et al., 2019; Jacobs & Wallach, 2021). Additionally, transparency is also answered by means of explainability and interpretability, responsible democratization, transparent model reporting, benchmarked evaluation in a variety of conditions, such as across different cultural, demographic and intersectional groups that are relevant to the intended application domains (Mitchell et al., 2019).

**Our Contribution:** We propose a framework "Design for Trust" where we demonstrate (i) how historical data biasness is identified, mitigated and risk assessments are performed (ii) how to build robust model with feature sensitivity based cost function; perform various conformity assessment on the model and provide explainability (iii) how human interventions can be applied to adhere to regulatory and business needs and at the same time, flag potential risk or financial implications.

## 2 PROBLEM SETUP & SOLUTION METHODOLOGY

In this paper we have considered binary classification setting where training data points are represented as $(X, A, Y)$, where $X \in X$, is a feature vector, $A \in A$, is attribute, and $Y \in 0,1$, is a label. We have considered classifier, $H$, as a model to spot historical biasness in the data and classifier, $C$, as a model for conformity assessment.

### 2.1 DATA INTERPRETATION:

In order to bring transparency in data we propose; **Bias Parallel** – identify and mitigate historical biased decisions; **Data Imbalance** - flag lesser representative samples and rebalance through sampling; **Data Risk Assessment** - sub-group assessment and data insights. Classifier $H$ , in this stage has objective to learn and identify pattern of historic biasness.

**Bias Parallel**   Here we flag data points along with corresponding features and proxy features resulting in historical biased decision. For instance, if decision gets changed on basis of Gender, we flag the data point as potential bias. Using the classifier, $H$, we simulate various decision changes based on feature attribute change and flag the data points. In case of continuous values we performed discretization and simulated the same. Bias mitigation is performed via delete and rectify operation on actual data point. We highly insist on performing these operations in conjunction with domain expert as it helps in identifying root cause behind the biasness and also identifying if any of the potential reasons or features were missed out in initial stages. Given loan status for $H(x)$, where $x$ represents vector associated with applicant and $[A1, A2] \in [A]$ represents attributes associated with feature. We flag biased points based on condition:

$$H(LoanStatus|\ x\ where\ \text{A} = \text{A1})! = H(LoanStatus|\ x\ where\ \text{A} = \text{A2}) \tag{1}$$

**Data Imbalance**   Dataset with improper sub-group representation sample can result in introducing biasness in the model and so rebalancing the data set plays a crucial role. Firstly, we calculate the percentage of attribute space corresponding to each feature and flag it if the imbalance exceeds certain defined threshold. Secondly, we build a decision tree on the dataset and as we go depth wise we see the representation of each of the strata and flag the strata where False Positive Rate **(FPR)** and False Negative Rate **(FNR)**, derived from the classification model $H$, exceeds beyond certain defined threshold (Pastor et al., 2010). To perform rebalancing we used both oversampling as well as synthetic data generation based on the property of the identified strata.

**Data Risk Assessment**   Objective here is to provide insights on the historical decisions and flag potential risks in data. In order to provide insights, we use Shapely explainers, (Lundberg & Lee, 2017), on the trained classifier $H$. Using feature importance plot from the explainer, we are able to visualize ranked list of features which played important role in historical decision making and using summary plot from the explainer, we are able to showcase how the variation in values of each of the feature impacts decision making. We are also able to identify certain sensitive attributes like gender, age, marital status and proxy features like "Has Telephone" impact in decision making and flag them as potential risks. Summary plot also helps us in identifying certain corner cases where the impact is different from rest of the points, potentially due to being a completely unique combination of attribute values.

### 2.2 CONFIRMITY ASSESSMENT:

In this stage we build a robust model adhering to both business and regulatory needs, perform various tests to validate the robustness of the model and provide explainability on model decisions.

**Cost Function**   We propose a sensitivity based cost function where the adversarial samples generated based on identified sensitive data points during the training process are considered to minimize the impact of sensitive features on the model outcomes.

$$TotalLoss = BinaryCrossEntropyLoss(BCE) + SensitivityLoss(SE) \tag{2}$$

$$BinaryCrossEntropyLoss(BCE) = \frac{1}{n}\sum_{i=1}^{n}(y_i * log(y_i^{prob}) + (1 - y_i) * log(1 - y_i^{prob})) \tag{3}$$

$$SensitiveLoss(SE) = k * [P(x) - P(x')] \tag{4}$$

$$sensitivity = \frac{\partial}{\partial x}(BinaryCrossEntropyLoss) \tag{5}$$

where $k$: regularization parameter; $x$: actual data point; $x'$: perturbed data point; $P(x)$: probability score for $x$; $P(x')$:probability score for perturbed input $x'$;

Sensitive loss function minimize the effect of sensitive features in model outcomes via applying penalty for highly sensitive data points. To derive perturb input $x'$, we calculate the gradient of $BCE$ loss with respect to the input $x'$, equation (5). To identify the sensitive data point based on identified feature from data interpretation stage, we condition a threshold, $\boldsymbol{T}$, such that we select only those data points which satisfies the criteria **abs(sensitivity) $> \boldsymbol{T}$**, conditioned over sensitive feature attributes. We perform perturbation on the sensitive features of the identified data points $x$ to generate $x'$. The model weights are updated based on calculating gradient of total loss with respect to the model weights. The perturbed data points $x'$, serves as adversarial samples during training process and resulted in improving the robustness. The number of sensitive data points reduced significantly compared to model where loss function has no sensitivity loss component. Additionally, it was also identified that points close to decision boundary were only the sensitive points flagged during adversarial test phase. Whereas, earlier many points probability score shifted significantly on flipping the attribute values identified during what-if-analysis.

**Algorithmic Assessment**    Once we have classifier, $C$, available from the optimized Cost Function, we perform various assessments on the model to test its robustness.

CONCEPT TEST    In this test, we try to evaluate how various concepts affect decision making, where concept refers to feature or feature combinations under test (Kim et al., 2018). For instance, how important is Gender as concept in decision making. Here we also try to create comprehensive explanation for the end users to make it easy to interpret. Combination of features can be consolidated to represent income, assets and liabilities of an applicant which make results more interpretable. For instance, in loan approval case various features like salary, job class, employment status etc. falls as a part of income. In order to get score corresponding to each of the feature, we fixed the neural architecture at second last layer with number of neurons equivalent to that of number of input features. The value derived from the normalized activations in this layer served as score for evaluating concept score.

WHAT-IF-ANALYSIS    In this test we perform interventions over various attributes contributing to decision making and validate how each of the attributes impacts model output (Wexler et al., 2019). The feature interventions for evaluation are fixed with box constraints so that realistic business scenarios can be validated. Interventions showing significant changes in the model output are analyzed carefully as they may be potential sensitive points and are flagged immediately.

CASE SPECIFIC EXPLAINABILITY    Here we showcase explainability for each of the model decisions. We use the activation scores from the bottleneck layer to provide explainability for each of the feature in decision making. For providing actionability, we find data point $x'$, where $x' \in X$, such that we minimize the distance between actual data point $x$ and $x'$, subject to outcome of classifier, $C(x')$, being the desired class output. Equation (6) below encourages counterfactual to be close to desired class and equation (7) encourages data points to be close to original data point (Verma et al., 2020).

$$arg \min_{x'} d(x, x'); \qquad subject\ to\ C(x') = y' \tag{6}$$

$$arg \min_{x'} \max_{\lambda} \lambda * [C(x') - y']^2 + d(x, x') \tag{7}$$

GENERATING ADVERSARIAL SAMPLES    Objective of this test is to perform model robustness check via generating perturbed samples where the classification decision are sensitive i.e. decision class changes on making small perturbation to original data point (Carlini & Wagner, 2017; Cartella et al., 2021). The adversarial perturbations created were subjected to box-constraints as shown in equation (9) in order to make sure perturbed values does not fall outside the desired range, where

Table 1: Detected biased points using AI model to spot historic biasness

| | Model_V1 | | | Model_V2 | | |
|---|---|---|---|---|---|---|
| **AUC Score** | 0.74 | | | 0.82 | | |
| **Detected BIAS points** | **Gender** | **Age** | **Marital Status** | **Gender** | **Age** | **Marital Status** |
| | 87 | 54 | 20 | 139 | 54 | 58 |
| **Generated Adversarial Samples** | 84 | | | 113 | | |

$box_{max}$ represents feature vector $max(x) \in X$ and $box_{min}$ represents feature vector $min(x) \in X$;

$$optimization\ function = \min_{x'} \begin{cases} d(x, x') + \lambda * max(C(x') - \Theta, k) & if\ y = 1, \\ d(x, x') + \lambda * max(\Theta - C(x'), k) & if\ y = 0. \end{cases} \tag{8}$$

$$x' = \frac{1}{2}\{tanh(x') * [box_{max} - box_{min}] + [box_{max} + boxmin]\} \tag{9}$$

where $\Theta$: decision boundary threshold, $k$: confidence score, $d(x, x')$: distance between $x \& x'$.

We evaluated results on both with and without sensitive loss in cost function and found that with sensitive loss function, the number of sensitive points reduced significantly. In fact, most of the data points detected by adversarial generation algorithm were the ones close to decision boundary.

### 2.3 DECISION INTELLIGENCE:

Objective of this stage is to provide reviewers the ability to evaluate and modify AI decisions in cases that requires special support based on business and regulatory requirements. Firstly, possibility for human intervention where the users will be able to select the specific features and shift the decision boundary. This would help in identifying the change specific to the selected sub-group in terms of count, % change in count and compare improvement with the past results. Secondly, impact analysis, where users will be able to view insights on financial impact and potential risks.

## 3 EXPERIMENTS AND RESULTS:

In this section we describe the experiments performed on the German Credit Dataset that classifies people described by a set of attributes as good or bad credit risks and showcase obtained results. The data consists of 29 features with 7 integer and 22 categorical ones. The strategy described is applied to other datasets as well but are not detailed due to space limitations. Also we showcase our results only on the loan category - "Business", however the results were generated for each of the loan category and evaluated.

**Model Construction** We used Neural Network with 3 hidden layers and the output layer constructed with a single neuron and sigmoid activation function to provide probability score, $P \in [0, 1]$. The last hidden layer was constructed with 2 neurons to plot Decision intelligence in 2-D plane representing all the data points and decision boundary. The second last hidden layer was constructed with number of neurons equal to feature inputs to the model (for deriving Concept Test scores). In case of "Data Interpretation" we used $BCE$ as loss function, while in case of building "Algorithm for trust" for "Conformity Assessment" we used $BCE$ along with Sensitivity Loss. Optimal thresholds were obtained using F1 score maximization as target metric. The model after undergoing "Conformity Assessment" was used for "Decision Intelligence".

**Results** Trained Classification model,$H$, was used to flag biased decision points. Table 1, shows comparison results for models. Model_V1 was the first version of the model with AUC score of 0.74. From the diagnostics report, we identified lesser subgroup representation and oversampled data from the identified strata, resulting in improved performance Model_V2 (AUC: 0.82). As the model's ability to capture historic pattern improves, the ability to detect bias points also improves simultaneously. Additionally, increase in generated adversarial data points is evident of the fact that when the model is trained on biased data, where objective is performing with higher accuracy, the model becomes more sensitive and hence bias treatment is absolute necessity. The plot in Figure 1(a) represents feature importance based on which decisions were made in the past. Clearly

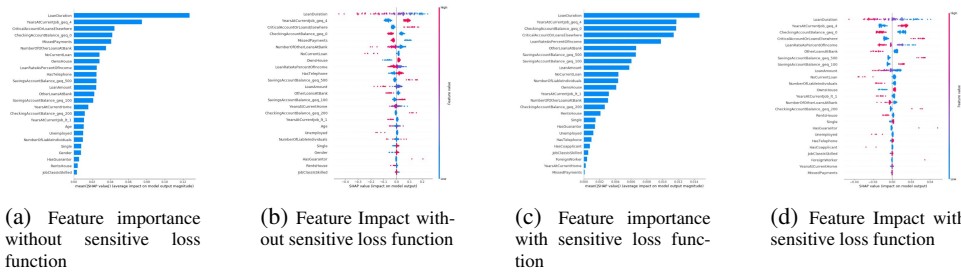

(a) Feature importance without sensitive loss function

(b) Feature Impact without sensitive loss function

(c) Feature importance with sensitive loss function

(d) Feature Impact with sensitive loss function

Figure 1: Model Insights.

Table 2: Sensitive data points count and variation with and without sensitive loss component in loss function

|  | Gender | | Age | | Marital Status | |
|---|---|---|---|---|---|---|
|  | Without Sensitive Loss | With Sensitive Loss | Without Sensitive Loss | With Sensitive Loss | Without Sensitive Loss | With Sensitive Loss |
| Sensitive Points | 87 | 6 | 54 | 0 | 20 | 14 |
| Prediction Score Variation | 0.16-0.69 | 0.489-0.491 | 0.21-0.93 | NA | 0.39-0.55 | 0.486-0.492 |
| Standard Deviation | 0.084 | 0.0003 | 0.148 | NA | 0.027 | 0.0002 |

we can see that sensitive features Gender, Age and Single (Marital Status) are one of the contributing factors for decision making. If we look deeper into summary plot in Figure 1(b), we can see their impact in decision making. At the end of data interpretation stage, we have (a) clear understanding of data in terms of biasness it possess (b) features which might be sensitive and is influencing decision making. The biasness in the flagged data points are mitigated in conjunction with domain experts and the data is ready for building trustworthy model.

While building model for "Confirmity Assessment", we minimize impact of sensitive features Gender, Age and Marital Status on decision making via training on sensitivity based Cost Function. It can be observed from Figure 1, both feature importance as well as their impact on the sensitive features has been reduced drastically. Also from the Table 2, we can see that on addition of sensitive loss function, the count as well as the variation of score for sensitive features has been reduced significantly. It was also noticed that the sensitive data points detected with sensitive loss function were very close to the decision boundary, which is an indication of the fact that the sensitivity with respect to features Gender, Age and Marital Status has been neutralized to greater extent.

In Figure 2, we showcase how important is income, asset and liability as a concept in decision making based on the score obtained from Concept Test. Applicants with high income, quality asset and low liability are more likely to get loans approved. However, as the income and quality of asset falls and liability increases, the applicant's loan is more likely to get rejected. Feature score indicators such as score for applicant loan amount, loan duration, missed payments asked for etc. adds to case specific explainability. A what-if-analysis interface along with counterfactual explanations supports explanation for decision made by the algorithm and fairness in terms of sensitive attributes.

Once the algorithm fairness and robustness was evaluated, we used the model for impact analysis by performing intervention in Decision Intelligence stage. In Figure 3. we can see projection of the data points along with the decision boundary with threshold 0.49. Based on the business and regulatory needs, the interface used provides flexibility to shift decision boundary (0.475 in Figure 3.) depending on the feature attribute and flag potential financial risk. In the Figure 3. above we showcase intervention for Gender, where threshold for both Male and Female can be customized and accordingly impact can be analyzed.

## 4 CONCLUSION

In this paper we presented, how to build a trustworthy, reliable, unbiased and robust AI solution via developing trust in the data used by capturing historical bias in decisions, and providing insights on historic decisions; trust in the algorithm via sensitivity based cost function and conformity assessments; and, finally, decision intelligence for human intervention, based on business and regulatory needs and provide insights on financial impact and potential risks. In the future, we would like to extend this framework towards different industry specific contextualization.

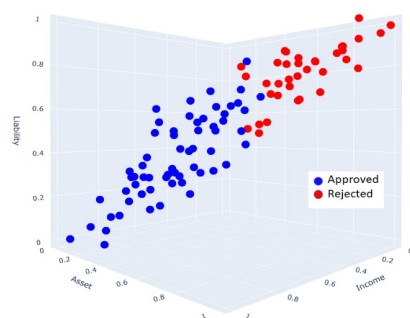

Figure 2: Concept test: Income, Asset and Liability as concepts for decision making.

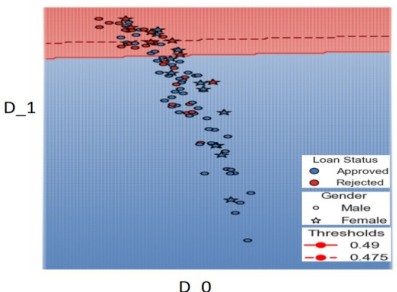

Figure 3: Decision Intelligence: Projection of data and decision boundary in 2D space.

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
