# OpenReview forum: "Design For Trustworthy AI Solutions"
_ICLR.cc/2022/Workshop/OSC — Submitted to ICLR2022 OSC _

### Official Review · Reviewer_j7R7 · 2022-03-12
**DESIGN FOR TRUSTWORTHY AI SOLUTIONS**

**Rating:** 1
**Confidence:** 3

**Review:**

Summary
This paper proposes a framework for identifying and debiasing AI: 1) data interpretation 2) conformity assessment 3) decision intelligence. The authors describe all steps and provide objectives of equations for several, and then apply this to a German credit dataset to debias loan predictions with respect to several sensitive features.

Pros
-Nice cost function and use of generated adversarial samples
-SHAP analysis

Cons
-This paper is good, but is not related to this workshop. This workshop is on objects, structured approaches, causality, and downstream applications such as RL, planning, control, and robotics. While there is need to consider societal and ethical implications of both the technical details and downstream applications of these techniques, this paper does not consider the kinds of models, problems, and data used in this area, and instead considers a binary classifier. Major AI conferences like NeurIPS, ICML, and ICLR usually have several workshops on ethics, fairness, and interpretability each year. If this paper is not accepted to this workshop, I would recommend targeting one of those workshops in the future, as they will be able to provide better feedback and learning opportunities.
-Broad concepts of studying feature importance, debiasing, removing sensative features, etc. are already well known in fairness community
-More interpretation of SHAP values would be informative
-When is a loan decision fair or ethical?

Notes:
 -"(Hellstrom¨ et al., 2020)" reference on line 3 should be Hellström
-Figure 1: too small, hard to read
-Figure 2: 3D scatterplot is hard to interpret in 2D
-Figure 3: hard to determine what gender balance is
-Focusing more on the German credit dataset could help set this paper apart from past work and make motivation clearer

---

### Official Review · Reviewer_52yC · 2022-03-15
**Unclear contribution, mostly a collection of well-known heuristics, possibly also off-topic**

**Rating:** 1
**Confidence:** 3

**Review:**

The paper proposes "Design for Trust", a conceptual framework for addressing transparency in AI solutions, composed by a Data interpretation, a Conformity Assessment and Decision intelligence phase.

The paper doesn't seem to provide any technical solution, but mostly heuristics based on having oracle classifiers for bias and conformity,  so that one can simulate various decisions and create new data. Given these oracles and under the supervision of an expert, one can delete and rectify the decision in the data, so it's not biased.

In general the paper is not very well-written and the contributions are not very clear, besides describing a set of heuristics or applications of existing methods. Also I have the impression that it is a bit off-topic, since it seems to focus on trustworthiness of AI applications, which I think is not too related to the workshop aim.

Minor details:
- several typos with singular vs plural nouns or missing an article, e.g. "we test trustworthiness, robustness and explainability of AI algorithm" or "Classifier H , in this stage has objective to learn and identify pattern of historic biasness."
- "Shapely explainers"

---

### Decision · Program_Chairs · 2022-03-21

**Decision:**

Reject

**Comment:**

Unfortunately both reviewers agreed that the paper is not ready for presentation. I recommend the authors to look at the reviews and use the feedback to improve the paper.